# Whole Tumor Histogram Analysis Using DW MRI in Primary Central Nervous System Lymphoma Correlates with Tumor Biomarkers and Outcome

**DOI:** 10.3390/cancers11101506

**Published:** 2019-10-08

**Authors:** Insun Chong, Quinn Ostrom, Bilal Khan, Dima Dandachi, Naveen Garg, Aikaterini Kotrotsou, Rivka Colen, Fanny Morón

**Affiliations:** 1Department of Radiology, Baylor College of Medicine, Houston, TX 77030, USA; insunchong@icloud.com (I.C.); bilal.khan@bcm.edu (B.K.); 2Section of Epidemiology and Population Sciences, Department of Medicine, Dan L Duncan Comprehensive Cancer Center, Baylor College of Medicine, Houston, TX 77030, USA; quinn.ostrom@bcm.edu; 3Division of Infectious Disease, Department of Medicine, University of Missouri—Columbia, Columbia, MO 65212, USA; dandachid@health.missouri.edu; 4Department of Diagnostic Radiology, The University of Texas MD Anderson Cancer Center, Houston, TX 77030, USA; ngarg@mdanderson.org (N.G.); rcolen@mdanderson.org (R.C.); 5Department of Cancer Systems Imaging, The University of Texas MD Anderson Cancer Center, Houston, TX 77030, USA

**Keywords:** primary central nervous system lymphoma, magnetic resonance imaging, diffusion-weighted imaging, apparent diffusion coefficient, patients living with HIV, Ki-67

## Abstract

The ability to non-invasively predict outcomes and monitor treatment response in primary central nervous system lymphoma (PCNSL) is important as treatment regimens are constantly being trialed. The aim of this study was to assess the validity of using apparent diffusion coefficient (ADC) histogram values to predict Ki-67 expression, a tumor proliferation marker, and patient outcomes in PCNSL in both immunocompetent patients and patients living with HIV (PLWH). Qualitative PCNSL magnetic resonance imaging (MRI) characteristics from 93 patients (23 PLWH and 70 immunocompetent) were analyzed, and whole tumor segmentation was performed on the ADC maps. Quantitative histogram analyses of the segmentations were calculated. These measures were compared to PCNSL Ki-67 expression. Progression-free survival (PFS) and overall survival (OS) were analyzed via comparison to the International Primary Central Nervous System Lymphoma Collaboration Group Response Criteria. Associations between ADC measures and clinical outcomes were assessed using univariate and multivariate Cox proportional hazards models. Normalized ADC (nADC)_Min_, nADC_Mean_, nADC_1_, nADC_5_, and nADC_15_ values were significantly associated with a poorer OS. nADC_Max_, nADC_Mean_, nADC_5_, nADC_15_, nADC_75_, nADC_95_, nADC_99_ inversely correlated with Ki-67 expression. OS was also significantly associated with lesion hemorrhage. PFS was not significantly associated with ADC values but with lesion hemorrhage. ADC histogram values and related parameters can predict the degree of tumor proliferation and patient outcomes for primary central nervous system lymphoma patients and in both immunocompetent patients and patients living with HIV.

## 1. Introduction

Primary central nervous system lymphoma (PCNSL) represents a rare, aggressive subgroup of Non-Hodgkin Lymphoma [1,2]. Although the majority of the cases of PCNSL are sporadic, a minority are associated with immunosuppression, such as in patients living with HIV (PLWH) [1,2,3]. Treatment regimens for PCNSL remain restricted due to the limited number of agents that can penetrate the blood-brain barrier. Development of an ideal treatment regimen for PCNSL is complex as PCNSL in PLWH is a distinct entity from other central nervous system (CNS) neoplasms as it is associated with the Epstein-Barr virus, which allows for the additional possibility of antiviral-based treatment regimens [1,2,3,4]. 

Ki-67 is a proliferation marker that has been evaluated in lymphomas and other malignancies [5,6,7]. Some studies have shown that the proportion of malignant cells positively stained for Ki-67 may have prognostic importance in PCNSL and correlate with clinical outcomes [8,9].

Identifying reliable methods of non-invasively predicting outcomes is an important area of research as new treatment regimens in both immunocompetent patients and PLWH are constantly being investigated [10,11,12,13,14]. Magnetic resonance imaging (MRI) is the modality of choice for imaging PCNSL, although definitive diagnosis requires histology [15,16,17,18]. Diffusion weighted imaging (DWI) and corresponding apparent diffusion coefficient (ADC) maps can reflect the microscopic cellular environment [10,19]. ADC histogram values have been reported to predict tumor cellularity in a broad range of neoplasms including lymphomas [10,14,19]. Furthermore, ADC values have also been shown to correspond to tumor markers such as Ki-67, predict survival, and serve as an effective means of monitoring treatment response in PCNSL [12,13,20,21,22,23]; however, these prior studies were limited by smaller sample sizes, using single regions of interests to obtain ADC values rather than using more accurate whole tumor segmentation, and the absence of data on PLWH, a patient population classically affected by PCNSL [22,24].

Therefore, our goal was to describe and compare the imaging findings of PCNSL for PLWH and immunocompetent patients, the relationship between Ki-67 expression and ADC values, and the relationship between prognosis and ADC values.

## 2. Results

### 2.1. Patient Demographics 

We retrospectively studied patients with PCNSL who were >18-years-old and had brain parenchymal PCNSL that presented to The University of Texas MD Anderson Cancer Center (MDACC) and Ben Taub Hospital (BTH) during the study period, who had no evidence of systemic lymphoma by whole-body computed tomography or positron emission tomography scan, and bone marrow biopsy. We excluded patients with diffuse lymphoma with CNS involvement or a relapse in the CNS, patients without pathology-proven PCNSL, and patients who had no preoperative brain MRI or had suboptimal imaging. Subsequently, we identified 93 patients, of which 32 (34.4%) were from BTH and 61 (65.6%) were from MDACC. Forty-nine patients (52.7%) were non-Hispanic/white, 21 (22.6%) were African American, 20 (21.5%) were Hispanic and 3 (3.2%) were Asian. Twenty-three patients (*24.7%*) were human immunodeficiency virus (HIV)-positive and 70 (75.3%) were HIV-negative. Of the African American patients, 65% were HIV-positive. Further patient demographic information is summarized in Table 1 and Table 2.

### 2.2. Qualitative Imaging Characteristics

Qualitative imaging characteristics were reviewed by one of the authors (F.E.M.) who is a board-certified neuroradiologist. The presence of ring enhancement was higher for PLWH (56.5%) compared to the HIV-negative subset (4.3%), which was statistically significant (Table 2, *p* < 0.001). Additionally, the presence of multiple lesions was found to be higher in the PLWH population (73.9%) compared to the HIV-negative subset (47.1%), which was also statistically significant (*p* = 0.046). Lesion macrohemorrhage, identified as areas of gradient-recalled echo (GRE) or susceptibility-weighted imaging (SWI) hypointensity and/or T1 shortening on the non-contrast T1-weighted imaging, was associated with HIV status (*p* = 0.047) as PLWH had a lesion hemorrhage rate of 69.6% versus 42.9% in patients with a HIV-negative status. 

### 2.3. Correlations between ADC Values and Ki-67 Expression

Ki-67 expression of lesions were determined via histopathological analyses of samples obtained by biopsy, partial resection, or complete resection (percentages provided in Table 2). Apparent diffusion coefficient (ADC) values were obtained by performing whole tumor segmentation (Figure 1) using 3D Slicer (version 4.7, SlicerSolutions, Boston, MA, United States), which was done by one of the authors (I.C.) [25]. If a patient had multiple lesions, the largest lesion was used for segmentation and analysis. If applicable, lesion hemorrhage was also excluded prior to segmentation. These segmentation volumes of interest (VOIs) were reviewed by two board-certified neuroradiologists (F.E.M. and R.R.C.) in consensus (i.e., simultaneously). Whole tumor segmentation was performed to allow for the evaluation of the entire hypercellular volume of interest’s intra-tumor heterogeneity, rather than a single slice region of interest placement (summary of the mean values of tumor ADC histogram parameters provided in Appendix A) [22]. Among the subset of patients with available Ki-67 expression data (28 of 93), the relationships between Ki-67 expression and ADC parameters were explored. As compared to the larger dataset, these patients were largely from MDACC (92.9%), HIV− (92.9%), older (60.7% over 60 years old), and received biopsy (78.6%, as compared to 17.9% with partial resection and 3.6% with complete resection). Various ADC parameters extracted from the segmented lesions were found to have a statistically significant correlation with Ki-67 expression (Table 3). In all patients (*n* = 28), no significant correlations were identified. However, when only lesions without hemorrhage were included (*n* = 18), nine absolute ADC parameters were identified to have positive significant correlation to Ki-67 expression, which are as follows: ADC_Min_ (*p* = 0.01), ADC_Mean_ (*p* = 0.02), ADC_1_ (*p* = 0.01), ADC_5_ (*p* = 0.01), ADC_15_ (*p* = 0.02), ADC_16_ (*p* = 0.02), ADC_75_ (p = 0.03), ADC_95_ (*p* = 0.04), and ADC_99_ (*p* = 0.045) (Appendix A). When values were normalized using white matter values (nADC), there were no significant associations between ADC values and Ki-67 expression, although the values correlated inversely (Table 3).

### 2.4. ADC Values and Overall Survival 

ADC values were also found to have relationships with poorer overall survival (OS) when adjusted for age, HIV status, Eastern Cooperative Oncology Group (ECOG) and treatment pattern (Table 4). Hazard ratios for OS in all patients revealed statistical significance for poorer OS with five ADC parameters generated by dividing the tumor value against the corresponding normal white matter value to correct for variations in imaging technologies within the dataset (nADC), which are as follows: nADC_Min_ (*p* = 0.02, Hazard Ratio (HR) = 0.532, 95%CI = 0.294–0.963), nADC_Mean_ (*p* = 0.048, HR = 0.689, 95%CI = 0.395–1.199), nADC_1_ (*p* = 0.006, HR = 0.500, 95%CI = 0.275–0.907), and nADC _5_ (*p* = 0.02, HR = 0.559, 95%CI = 0.314–0.996), and nADC_15_ (*p* = 0.03, HR = 0.717, 95%CI = 0.409–1.257). ECOG status and treatment pattern had a statistically significant effect on OS in all models. When lesions with hemorrhage were excluded, the ADC_Min_, nADC_Min_, nADC_1_, nADC_5,_ nADC1_5_ parameters were significantly associated with poorer survival, while ECOG and treatment pattern had independent statistically significant effects. When analysis was stratified by HIV status, no ADC values were associated with OS in HIV+ patients, while higher values of two (nADC_Min_, nADC_Mean_) were associated with poorer survival and five (ADC_1_, nADC_5_ nADC_15_ nADC_75_ nADC_95_) were associated with improved survival in HIV− patients. ECOG status exerted a statistically significant effect on survival in HIV− patients only, but treatment significantly affected survival in both groups.

### 2.5. ADC Values and Progression-Free Survival 

There was a marginally significant association between nADC_min_ and progression free survival (PFS) (*p* = 0.05, HR = 0.602, 95%CI = 0.344–1.053). This association was statistically significant in HIV− patients (*p* = 0.03, HR = 0.557, 95%CI = 0.295–1.052) and marginally significant in HIV+ patients (*p* = 0.06, HR = 0.328, 95%CI = 0.067–1.603), and treatment pattern was statistically significant in both models (Table 5).

### 2.6. Additional Factors Impacting Overall Survival

Along with ADC values, additional factors, including lesion hemorrhage, HIV status, treatment with auto-stem cell transplantation, and ECOG scores impacted OS (Table 1 and Table 4). The absence of lesion hemorrhage was associated with better OS (*p* = 0.004). HIV status was associated with poorer OS, as the median survival time for PLWH was 6 months versus 40 months for HIV-negative patients (*p* = 0.02). Treatment with auto-stem cell transplantation was marginally associated with increased median OS with 76 months, compared to 37 months in patients who were not treated with stem cell transplantation. Lastly, ECOG scores between 2 and 4 had a median OS of 7 months versus 47 months for ECOG scores less than 2 (*p* < 0.001). Number of tumors, size of the largest tumor, tumor enhancement pattern, cortical invasion or deep brain involvement were not found to be associated with poorer survival outcomes. 

### 2.7. Additional Factors Impacting Progression-Free Survival

Additional factors, including lesion hemorrhage (*p* = 0.03) and ECOG scores (*p* = 0.02), impacted PFS (Table 1 and Table 5). Additionally, ECOG scores between 2 and 4 had median PFS of 5.5 months, which was significantly less than 24 months for ECOG scores less than 2 (*p* = 0.02). Periventricular location and ependymal involvement also impacted PFS in immunocompetent patients (*p* = 0.04, *p* = 0.03). Size of the largest tumor, number of lesions, tumor enhancement pattern or HIV status were not associated with poorer PFS. 

## 3. Discussion

The ability to non-invasively predict outcomes and monitor treatment response in primary central nervous system lymphoma (PCNSL) is important as new treatment regimens are constantly being trialed. We aimed to investigate how ADC histogram profiling with whole tumor segmentation reflects tumor proliferation and overall patient outcomes in both patients living with HIV and immunocompetent patients with PCNSL. Additionally, we aimed to investigate qualitative imaging characteristics of PCNSL to explore additional factors that may influence prognosis. This information may eventually allow both clinicians and radiologists to accurately analyze tumor proliferation and predict patient prognosis without the need for invasive procedures such as biopsies.

Prior studies have demonstrated significant correlations between ADC parameters and Ki-67 [6,7,12,13,20,21,22,23,26]. Our results with the normalized ADC histogram values demonstrate these same relationships but were not statistically significant. This may be due, in part, to tumor hemorrhage, and though macrohemorrhage was excluded from our lesion segmentation, microhemorrhage could not be accounted for. Tumors with hemorrhage (46 of 93 patients; 16 of 23 PLWH) have blood products that age heterogeneously based on contributions from the regional oxygen tension, the concentration of tissue macrophages, and the presence of tumor cells [27]. These factors result in differing levels of contributions to ADC signal from T2-relaxation, T1-relaxation, and T2*-effects, thereby resulting in an alteration of the relationship between regional cellularity and ADC [28]. Another consideration is that hemorrhage results in local cellular heterogeneity, which can cause changes in the correlations between ADC and Ki-67 [21,29]. Additionally, tumor necrosis may not have been optimally excluded as the ADC maps were not co-registered with T1 post-contrast images, which may be a future area for further research.

Our findings agree with prior studies that demonstrated significant associations between several ADC parameters and poorer OS when lesions with hemorrhage were excluded but also found an association between nADC_15_ and poorer OS [12,20,21,22,23]. Our results also agree with prior studies examining qualitative PCNSL imaging characteristics in PLWH that found a higher rate of ring enhancing lesions and lesion hemorrhage in PLWH [18,30,31]. The lack of significant associations between PCNSL ADC parameters and survival in PLWH may be due to two factors: 1) distortion of the ADC profile of the tumor due to intratumor hemorrhage resulting in alterations of the tumor microenvironment as previously stated and 2) inter-/intra-tumor variability with PCNSL in PLWH, due in part to lesion necrosis [16,18,27,28,29,32]. However, lesion hemorrhage was significantly associated with OS and PFS in both immunocompetent patients and PLWH [33]. Ependymal involvement and periventricular location, found to be characteristic of PCNSL, were also significantly associated with PFS in immunocompetent patients [30,31]. Relationships between hemorrhage, ependymal involvement, and periventricular location to outcomes have not been well-established in the PCNSL, indicating possible separate prognostic factors to consider in patients with PCNSL and additional control measures to be used for further research purposes [33]. 

However, unlike prior studies on PCNSL histogram analysis, our study further substantiates the expanding literature on the relationship between ADC values, patient outcomes, and tumor pathologic findings by (1) utilizing a larger patient population, (2) performing whole tumor segmentation rather than using single region of interest for ADC analysis, (3) performing exclusion analysis for lesion hemorrhage and necrosis, and (4) including PLWH, an important patient population affected by PCNSL. Our hope with this work is to motivate additional studies with perhaps even larger cohorts and increase validations of these relationships in order to automate the analyses of tumor proliferation and patient outcomes via integration with either a picture archiving and communication system (PACS) or a radiology informatics system and to facilitate the creation of predictive models which can be incorporated into clinical support systems.

### Limitations

There are several limitations with this study. Firstly, patients were selected from only two institutions, both in the same city. Secondly, the patients were scanned using several different machines, although this confounding effect was minimized by using ADC values of normal white matter from the side contralateral to the tumor, a method that has been previously reported [10,12,23]. Thirdly, the degree of patient exclusion was a limitation of this study. While the sample size of the study was relatively larger than prior studies, lesions with hemorrhage were further excluded for portions of the analysis. This presents a potential limitation in using ADC histogram analysis for patients with PCNSL complicated by lesion hemorrhage. Lastly, the biopsied portion of PCNSL specimen used to calculate Ki-67 expression may not be representative of its expression throughout the whole tumor and may decrease the accuracy of the statistical analysis.

## 4. Materials and Methods

### 4.1. Patient Selection and Review

We conducted a Baylor College of Medicine IRB-approved (H-39346, 27 February 2018), Health Insurance Portability and Accountability Act compliant (HIPAA) retrospective study of patients diagnosed with PCNSL at the University of Texas MD Anderson Cancer Center between March 2000 and July 2016 and at Ben Taub Hospital, between January 2012 and December 2016. All the patients that met the following criteria were included: (1) age > 18 years; (2) patients with brain parenchymal PCNSL that presented to MDACC and BTH during the study period, who had no evidence of systemic lymphoma by whole-body computed tomography or positron emission tomography scan and bone marrow biopsy. Exclusion criteria were (1) patients with diffuse lymphoma with CNS involvement or relapse in the CNS, (2) patients without pathology proven PCNSL, and (3) patients who had no preoperative brain MRI or had suboptimal preoperative imaging. We reviewed the medical records and collected information on socio-demographic characteristics (e.g., gender, race, ethnicity), HIV status, biopsy results, immunohistochemical staining (including Ki-67), and clinical outcomes. Due to the retrospective nature of this analysis, not all measures were available on all individuals. 

### 4.2. MR Imaging Protocols

All images were acquired within the clinical diagnostic parameters using either a 1.5 or 3.0 Tesla GE (Milwaukee, WI, USA) or Siemens (Erlangen, Germany) scanner with the corresponding head coils. Diffusion-weighted imaging (DWI) images were obtained with the following parameters using spin echo and spin echo planar sequences: (1) for BTH: minimum Echo time (TE), 8000 ms repetition time (TR), no flip angle, 5 mm slice thickness (27 slices), and 240 mm^2^ field of view (FOV) with diffusion-sensitizing gradients applied with b factors of 0 and 1000 s/mm^2^ and (2) for MDACC: 8.9 ms TE, 6600 ms TR, 5 mm slice thickness, and 230 mm^2^ FOV with diffusion-sensitizing gradients applied with b factors of 0, 500, and 1000 s/mm^2^. ADC maps were automatically generated by the operating console of the magnetic resonance (MR) scanner.

### 4.3. Analysis of MRI Imaging

For image analysis and segmentation, we used 3D Slicer (version 4.7, SlicerSolutions, Boston, MA, United States), an open source software platform for medical imaging informatics [25]. Images were exported from the picture archiving and communication system (PACS) in Digital Imaging and Communications in Medicine (DICOM) format and converted to a Neuroimaging Informatics Technology Initiative (NIfTI) file. The volume of interest (VOI) consisting of areas of ADC-restriction in a biopsy-proven tumor were outlined by one of the authors (I.C.) (Figure 1). If a patient had multiple lesions, the largest lesion was segmented and used for analysis. Regions of macrohemorrhage, identified as areas of gradient-recalled echo (GRE) or susceptibility-weighted imaging (SWI) hypointensity and/or T1 shortening on the non-contrast T1-weighted imaging, were excluded from the segmentation volumes. A normal white matter region of interest in the contralateral hemisphere was obtained for normalization purposes in order to account for any scanner variability [10,34]. The segmented images were reviewed in consensus (i.e., simultaneously) by two board-certified neuroradiologists (F.E.M. 15 years of experience, and R.R.C. 9 years of experience). 

Normalized ADC ratios were computed as the ratio of the ADC values within a lesion to the ADC values within normal white matter. Subsequently, 3D Slicer’s Label Statistics function was used to obtain minimum, maximum, mean, standard deviation, skewness, kurtosis, volume, and the percentile values (1st, 5th, 15th, 25th, 75th, 95th, and 99th) of the ADC map VOIs. Qualitative PCNSL characteristics, such as the locations of tumors, the presence of intra-tumor hemorrhage, and the enhancement characteristics on T1-weighted post-gadolinium contrast imaging of tumors, were reviewed by a board-certified neuroradiologist (F.E.M.).

### 4.4. Clinical Outcomes

Two clinical outcomes, overall survival and progression free survival, were analyzed. Complete remission was defined according to the International Primary Central Nervous System Lymphoma Collaboration Group Response Criteria [11,35]. Patients who did not receive treatment were excluded from this analysis. We calculated the OS from the time of diagnosis to death of any cause. 

### 4.5. Statistical Analysis

Comparisons of demographic data by HIV status was performed using chi-squared test and fishers exact test when any group contained five or less individuals. Median survival time was generated using Kaplan-Meier methods and *p*-values were calculated using Cox proportional hazards models. Outlier values in ADC mean from white matter images were detected using the Grubb’s test and four individuals were excluded from further analyses of imaging data. Normalized ADC ratios were generated by dividing the tumor value against the corresponding white matter value to correct for variations in imaging technologies within the dataset [10,12,23]. The association between qualitative imaging characteristics, OS, and PFS was compared using univariable and multivariable Cox proportional hazards models adjusted for age at diagnosis, ECOG score, HIV status, and treatment pattern. Treatment pattern was included as a categorical carriable using the following levels: supportive/palliative, whole brain radiation therapy, methotrexate monotherapy, methotrexate-based combination chemotherapy, and whole brain radiation therapy plus methotrexate-based combination chemotherapy. Statistical analyses were performed using R 3.5.0 [36]. Statistical significance was set as *p*-value < 0.05.

## 5. Conclusions

Our results show promise that ADC histogram values and tumor characteristics may be used by radiologists, after further research and confirmation, to predict the degree of tumor proliferation and patient survival in both immunocompetent patients with PCNSL and PCNSL in PLWH. As new treatment regimens for PCNSL are being trialed, it is important to add to the growing and exciting body of work that investigates the imaging findings of PCNSL, relationships between the imaging and the immunohistochemistry of PCNSL and the correlation between the imaging and patient outcomes. 

## Figures and Tables

**Figure 1 cancers-11-01506-f001:**
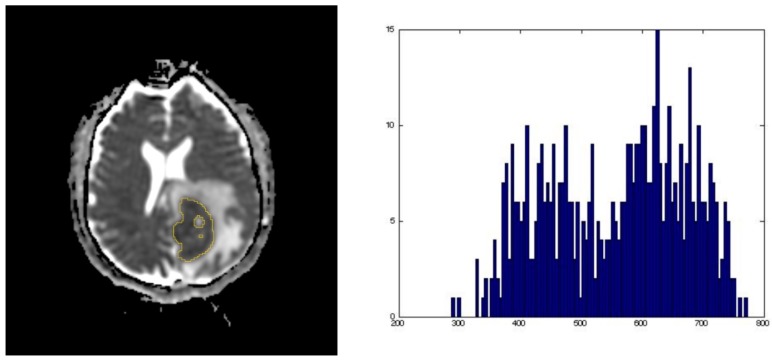
Representative images of an apparent diffusion coefficient (ADC) map whole tumor segmentation with its corresponding ADC histogram distribution used for data analysis.

**Table 1 cancers-11-01506-t001:** Patient information and qualitative imaging characteristics. Results are considered significant (*) when *p* < 0.05.

	All Patients (% of All Patients)	Deaths	Median Survival in Months (95% CI)	*p*-Value ^a^	Relapses	Median PFS in Months (95% CI)	*p*-Value ^b^
**Total**	93	55	41 (23–76)		57	11 (6–33)	
Study site							
BTH	32 (34.4%)	20	15 (5–**)	ref	23	6 (3–15)	ref
MDACC	61 (65.6%)	35	47 (37–1001)	0.04 *	34	15 (8-**)	0.04 *
**Gender**							
Male	51 (54.8%)	34	37 (11–52)	ref	34	7 (4–23)	ref
Female	42 (45.2%)	21	100 (16–**)	0.07	23	13 (8–**)	0.19
**Age**							
<60	50 (53.8%)	30	43 (15–**)	ref	31	11 (6–**)	ref
≥60	43 (46.2%)	25	43 (11–**)	0.75	26	11 (5–**)	0.82
**Race/Ethnicity**							
Non-Hispanic/White	49 (52.7%)	33	43 (11–76)	ref	31	11 (6–**)	ref
African-American	21 (22.6%)	14	9 (5–**)	0.46	12	6 (3–**)	0.90
Hispanics	20 (21.5%)	7	**	0.25	12	15 (10–**)	0.74
Asians	3 (3.2%)	1	16 (16–**)	0.67	2	6 (5–**)	0.86
**HIV status**							
Positive	23 (24.7%)	15	6 (2–**)	ref	15	6 (2–**)	ref
Negative	70 (75.3%)	40	45 (37–101)	0.02 *	42	11 (8–**)	0.26
**ECOG**							
0–1	54 (58.1%)	26	47 (43–**)	ref	29	24 (11–**)	ref
2–4	38 (40.9%)	28	7 (4–37)	<0.001 *	28	5.5 (2–14)	0.02 *
**Number of lesions**							
Single	43 (46.2%)	29	43 (24–76)	ref	29	8 (5–33)	ref
Multiple	50 (53.8%)	26	41 (9–**)	0.91	28	14 (6–**)	0.41
Location							
Deep brain	48 (51.6%)	25	41 (23–**)	ref	29	11 (6–**)	ref
Not deep brain	45 (48.4%)	30	43 (11–101)	0.74	28	11 (5–**)	0.93
**Hemorrhage**							
Yes	46 (49.5%)	32	15 (6–52)	ref	33	5.5 (4–15)	ref
No	47 (50.5%)	23	76 (41–**)	0.004 *	24	17 (11–**)	0.03 *
**Enhancement**							
None	1 (1.1%)	--	--	--	--	--	--
Ring	16 (17.2%)	11	5 (2–**)	ref	10	5 (2–**)	ref
Solid	76 (81.7%)	44	43 (27–100)	0.11	47	11 (7–36)	0.53
**Type of surgery**							
Biopsy	60 (65.9%)	35	43 (37–101)	ref	35	13 (6–**)	ref
Partial resection	22 (24.2%)	13	15 (9–**)	0.689	15	9.5 (6–**)	0.610
Complete resection	8 (8.8%)	4	24 (3–**)	0.863	4	11 (2–**)	0.912
Other	1 (1.1%)	1	16 (**–**)	0.446	1	5 (**–**)	0.359
**Initial Treatment**							
Supportive/Palliative	5 (5.4%)	5	1 (1–**)	ref	5	1 (1–**)	ref
Whole brain radiation (WBRT)	18 (19.4%)	11	24 (3-**)	<0.001 *	10	6 (3–**)	<0.001 *
MTX monotherapy	6 (6.5%)	6	6 (3–**)	0.008 *	6	2.5 (2–**)	0.048 *
MTX-based combination chemo	51 (54.8%)	28	47 (35-**)	<0.001 *	31	13 (9–**)	<0.001 *
WBRT and MTX based chemo	13 (14.0%)	5	100 (47–**)	<0.001 *	5	**	<0.001 *
**Stem Cell Transplant (SCT)**							
Auto-HSCT	9 (9.7%)	3	76 (47–**)	ref	3	**	ref
No SCT	84 (90.3%)	52	37 (11–53)	0.10	54	8 (5–15)	0.06

^a^ Significance of overall survival difference between the levels of characteristics; ^b^ Significance of progression-free survival difference between levels of characteristics. ** cannot be calculated. CI = confidence interval; BTH = Ben Taub Hospital; MDACC = MD Anderson Cancer Center; HIV = human immunodeficiency virus; ECOG = Eastern Cooperative Oncology Group; MTX = methotrexate; HSCT = Hematopoietic stem cell transplantation.

**Table 2 cancers-11-01506-t002:** Patient information and qualitative imaging characteristics comparing HIV-positive with HIV-negative patients. Results considered significant (*) when *p* < 0.05.

	HIV Positive (% of All HIV+ Patients)	HIV Negative (% of All HIV− Patients)	*p*-Value ^a^
**Total**	23 (25%)	70 (70%)	--
**Study site**			<0.001 *
BTH	22 (95.7%)	10 (14.3%)	
MDACC	1 (4.3%)	60 (85.7%)	
**Gender**			0.16
Male	16 (69.6%)	35 (50.0%)	
Female	7 (30.4%)	35 (50.0%)	
**Age**			**<0.001 ***
<60	23 (100.0%)	27 (38.6%)	
≥60	0 (0.0%)	43 (61.4%)	
**Race/Ethnicity**			**<0.001 ***
Non-Hispanic/White	2 (8.7%)	47 (67.1%)	
African-American	15 (65.2%)	6 (8.6%)	
Hispanics	6 (26.1%)	14 (20.0%)	
Asians	0 (0.0%)	3 (4.3%)	
**HIV status**			
Positive	--	--	
Negative	--	--	
**ECOG**			<0.001 *
0–1	4 (17.4%)	50 (71.4%)	
2–4	19 (82.6%)	19 (27.1%)	
**Number of lesions**			0.046 *
Single	6 (26.1%)	37 (52.9%)	
Multiple	17 (73.9%)	33 (47.1%)	
**Location**			0.86
Deep brain	11 (47.8%)	37 (52.9%)	
Not deep brain	12 (52.2%)	33 (47.1%)	
**Hemorrhage**			0.047 *
Yes	16 (69.6%)	30 (42.9%)	
No	7 (30.4%)	40 (57.1%)	
**Enhancement**			<0.001 *
None	1 (4.3%)	0 (0.0%)	
Ring	13 (56.5%)	3 (4.3%)	
Solid	9 (39.1%)	67 (95.7%)	
**Type of surgery**			1.00
Biopsy	15 (68.2%)	45 (65.2%)	
Partial resection	5 (22.7%)	17 (22.7%)	
Complete resection	2 (9.1%)	6 (8.7%)	
Other	0 (0.0%)	1 (1.4%)	
**Initial Treatment**			<0.001 *
Supportive/Palliative	3 (13.0%)	2 (2.9%)	
Whole brain radiation (WBRT)	16 (69.6%)	2 (2.9%)	
MTX monotherapy	0 (0.0%)	6 (8.6%)	
MTX-based combination chemo	4 (17.4%)	47 (67.1%)	
WBRT and MTX based chemo	0 (0.0%)	13 (18.6%)	
**Stem Cell Transplant (SCT)**			0.11
Auto-HSCT	0 (0.0%)	9 (12.9%)	
No SCT	23 (100.0%)	61 (87.1%)	

^a^*p*-value denotes statistically significant difference in characteristics between HIV− and HIV+_patients. BTH = Ben Taub Hospital; MDACC = MD Anderson Cancer Center; HIV = human immunodeficiency virus; ECOG = Eastern Cooperative Oncology Group; MTX = methotrexate; HSCT = Hematopoietic stem cell transplantation.

**Table 3 cancers-11-01506-t003:** Correlation with Ki-67 in patients with available data. Results are considered significant (*) when *p* < 0.05.

Parameter	All Patients (n = 28)		No Hemorrhage (n = 18)		HIV+ (n = 2)		HIV− (n = 26)	
	r	*p*-Value	r	*p*-Value	r	*p*-Value	r	*p*-Value
Skewness	0.151	0.45	0.050	0.84	--	--	0.128	0.53
Kurtosis	0.171	0.38	0.067	0.79	--	--	0.206	0.31
nADC_Min_	0.026	0.90	0.149	0.56	--	--	−0.001	0.99
nADC_Max_	−0.190	0.33	−0.255	0.31	--	--	−0.239	0.24
nADC_Mean_	−0.148	0.45	−0.058	0.82	--	--	−0.210	0.30
nADC_1_	0.032	0.87	0.155	0.54	--	--	−0.002	0.99
nADC_5_	−0.023	0.91	0.118	0.64	--	--	−0.052	0.80
nADC_15_	−0.105	0.60	0.027	0.92	--	--	−0.139	0.50
nADC_75_	−0.167	0.40	−0.097	0.70	--	--	−0.242	0.23
nADC_95_	−0.232	0.24	−0.268	0.28	--	--	−0.339	0.09
nADC_99_	−0.195	0.32	−0.232	0.35	--	--	−0.290	0.15

nADC = normalized apparent diffusion coefficient values.

**Table 4 cancers-11-01506-t004:** Hazard ratios for overall survival in all patients and patients without lesion hemorrhage. Results are considered significant (*) when *p* < 0.05.

Feature	All Patients (n = 93)		No Hemorrhage (n = 47)		HIV+ (n = 23)		HIV− (n = 70)	
	*p*-Value ^a^	HR ^b^ (95% CI)	*p*-Value ^a^	HR ^b^ (95% CI)	*p*-Value^a^	HR ^b^ (95% CI)	*p*-Value ^a^	HR ^b^ (95% CI)
Skewness	0.28	0.861 (0.468–1.585)	0.50	1.383 (0.485–3.946)	0.66	1.224 (0.288–5.203)	0.26	0.796 (0.406–1.561)
Kurtosis	0.38	1.443 (0.745–2.794)	0.82	1.703 (0.589–4.924)	0.03 *	2.742 (0.447–16.813)	0.56	0.848 (0.43–1.671)
nADC_Min_	0.02*	0.532 (0.294–0.963)	0.02 *	0.425 (0.143–1.265)	0.10	0.416 (0.09–1.925)	0.03 *	1.376 (0.704–2.688)
nADC_Max_	0.32	0.943 (0.54–1.649)	0.29	0.68 (0.25–1.848)	0.63	1.561 (0.474–5.136)	0.12	0.935 (0.469–1.864)
nADC_Mean_	0.048 *	0.689 (0.395–1.199)	0.07	0.3 (0.097–0.93)	0.61	1.067 (0.311–3.653)	0.03 *	1.384 (0.682-2.81)
nADC_1_	0.006 *	0.5 (0.275–0.907)	0.01 *	0.286 (0.087–0.945)	0.18	0.416 (0.09–1.925)	0.01 *	0.539 (0.274–1.061)
nADC_5_	0.02 *	0.559 (0.314–0.996)	0.02 *	0.307 (0.102–0.928)	0.44	0.578 (0.15–2.222)	0.02 *	0.674 (0.318–1.428)
nADC_15_	0.03 *	0.717 (0.409–1.257)	0.04 *	0.377 (0.12–1.184)	0.55	1.276 (0.394–4.13)	0.03 *	0.543 (0.271–1.086)
nADC_75_	0.08	0.599 (0.342–1.05)	0.09	0.318 (0.102–0.994)	0.69	0.721 (0.205–2.539)	0.03 *	0.507 (0.257–1.002)
nADC_95_	0.12	0.735 (0.424–1.272)	0.13	0.396 (0.125–1.255)	0.97	0.984 (0.287–3.369)	0.04 *	0.553 (0.277–1.104)
nADC_99_	0.26	0.801 (0.46–1.393)	0.25	0.478 (0.162–1.41)	0.66	1.476 (0.428–5.096)	0.06	0.552 (0.275–1.11)

Models adjusted for age at diagnosis, ECOG, HIV status (unless stratified by HIV status), and treatment pattern. ^a^
*p*-value is for continuous increase of value for all ADC values. ^b^ Hazard ratio is calculated by comparing below median to above median values for all ADC values. HR = Hazard Ratio. CI = Confidence Interval. nADC = normalized apparent diffusion coefficient values.

**Table 5 cancers-11-01506-t005:** Hazard ratios for progression-free survival in all patients and patients without lesion hemorrhage. Results are considered significant (*) when p < 0.05.

Feature	All Patients (n = 93)		No Hemorrhage (n = 47)		HIV+ (n = 23)		HIV− (n = 70)	
	*p*-Value ^a^	HR ^b^ (95% CI)	*p*-Value ^a^	HR ^b^ (95% CI)	*p*-Value ^a^	HR ^b^ (95% CI)	*p*-Value ^a^	HR ^b^ (95% CI)
Skewness	0.28	0.77 (0.43–1.379)	0.22	0.8 (0.31–2.067)	0.53	1.355 (0.301–6.097)	0.13	0.645 (0.337–1.234)
Kurtosis	0.14	1.071 (0.57–2.012)	0.63	0.565 (0.227–1.406)	0.03 *	2.585 (0.428–15.628)	0.34	1.015 (0.522–1.974)
nADC_Min_	0.05	0.602 (0.344–1.053)	0.18	0.479 (0.187–1.225)	0.06	0.328 (0.067–1.603)	0.03 *	0.557 (0.295–1.052)
nADC_Max_	0.93	1.273 (0.744–2.178)	0.54	1.258 (0.5–3.165)	0.71	1.311 (0.385–4.458)	0.44	1.016 (0.512–2.016)
nADC_Mean_	0.45	0.993 (0.588–1.677)	0.99	0.952 (0.382–2.373)	0.42	0.885 (0.246–3.179)	0.27	0.839 (0.45–1.566)
nADC_1_	0.13	0.634 (0.363–1.105)	0.30	0.526 (0.21–1.318)	0.10	0.328 (0.067–1.603)	0.14	0.593 (0.316–1.115)
nADC_5_	0.30	0.761 (0.445–1.3)	0.48	0.707 (0.298–1.679)	0.26	0.507 (0.123–2.097)	0.25	0.669 (0.357–1.251)
nADC_15_	0.41	0.912 (0.536–1.552)	0.60	0.793 (0.315–1.993)	0.33	1.158 (0.341-3.931)	0.30	0.697 (0.369–1.318)
nADC_75_	0.47	0.786 (0.464–1.332)	0.82	0.887 (0.348–2.263)	0.54	0.581 (0.154–2.186)	0.23	0.652 (0.346–1.228)
nADC_95_	0.59	0.919 (0.543–1.555)	0.58	1.163 (0.436–3.106)	0.89	0.82 (0.229–2.941)	0.24	0.745 (0.394–1.408)
nADC_99_	0.93	1.127 (0.663–1.916)	0.38	1.313 (0.505–3.412)	0.79	1.261 (0.353–4.499)	0.38	0.889 (0.467–1.694)

Models adjusted for age at diagnosis, ECOG, HIV status (unless stratified by HIV status), and treatment pattern. ^a^ The *p*-value is for continuous increase of value for all ADC values. ^b^ Hazard ratio is calculated by comparing below median to above median values for all ADC values. HR = Hazard Ratio. CI = Confidence Interval. nADC = normalized apparent diffusion coefficient values.

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
