# Peer review of "Whole Tumor Histogram Analysis Using DW MRI in Primary Central Nervous System Lymphoma Correlates with Tumor Biomarkers and Outcome"

_cancers, 2019, doi:10.3390/cancers11101506_

Round 1
Reviewer 1 Report
Additional questions:
Introduction. Reviewer agrees that a non-invasive imaging prediction is important but the manuscript introduction has to be improved. Please see below.
Line 49 . How does Epstein-Barr virus complicate treatment of PCNSL ? Please explain.
Line 51. In Ref.51 only de novo diffuse large B cell lymphoma was evaluated. Please correct sentence.
Line 52. Ref 6 and ref. 8 are the same. Please correct.
Line 53. CT and PET also can be used (Ref. 11, 12 from authors). Also quote from Ref.11 states "Ultimately, no imaging method can definitively diagnose PCNSL, and histology is required." Please reconsider statement that "(MRI) is the modality of choice for imaging PCNSL".
Results.
Main concern is a presence of p-values in a table.
Please explain the p-values calculated for Table 1. Specifically what does it mean p-value <0.001 for study site ? Compare to what imaging parameter ? What does it mean that "Age" younger and older than 60 years has <0.001 probability ? Between HIV positive and HIV negative ?
line 88. Can authors provide information about "qualitative imaging characteristics". What exactly protocol/guidelines were used for "hemorrhagge"identification?
Discussion:
Authors clam that: "Our results demonstrate that ADC histogram values have significant relationships with both Ki-67 and OS in patients with PCNSL." Although in line 91-92 they provide next statement:"When values were normalized using white matter 91 values, there were no significant associations between ADC values and Ki-67 expression." The only claimed significant correlation to ki-67 expression in cases without hemorrhage (line 88). Please explain.
Overall the the discussion is in a great dissonance with results.
For example authors use n=18 samples without hemorrhage and show correlation with ADC paramters. But in table 1 there are total 47 case with no hemorrhage? Why are authors changing the size of cohorts?
Limitations.
Authors are correctly identified all limitations.But using normaliziotion of ADC values by ADC values in white matter does not correct for use of different scanner with different B =1.5 and 3 T.
Methods.
Figure 1. The image looks like T2 WI image. Please check. The contour on image does not correspond contours. Please check. Or provide more representative image.
Overall there is is a big concern to how the statistics was done.
Conclusion is not supported by results.
Reviewer 2 Report
This paper is a description of using MRI parameters to predict outcome for PCNSL. There is a large amount of comparisons that have been performed, but the descriptions of these are poor. The “real” impact of these findings is not made very clear, thus despite a large amount of analyses, it is not obvious what real clinical impact these findings will have over already existing methods for predicting patient outcome, and/or in the selection of patients for clinical trial.
Major issues:
This paper appears to have been originally written for a different journal where the materials and methods section is read first. Instead, for this journal, the methods is placed after the discussion. Therefore, when read as is, the results section and study rationale is not easily understood. The manuscript needs to be re-written to move a lot of the content in the methods to earlier in the manuscript. Alternatively, the authors may want to consider submitting to a journal where the manuscripts are structured differently. This is an extremely difficult paper to read. There is minimal narrative to lead the reader through what was done, the findings and the implications of these results. It is more just a list of statistics. Addressing the point above should improve this. The percentages quoted in table 1 are very confusing, some relate to the total number, and some relate to the proportion of patients in each column, but this isn’t described well at all. It also isn’t obvious which parameters are being compared to calculate the p values, although once I read the methods at the end this made sense. This table needs to be reformatted or must be explained since the reader will not read the methods first. Why are there only 28 patients analysed in table 2, when it refers to “all patients”? were Ki67 values only available on 28 patients, and what are the demographics of this subgroup? There is no description or example shown of how Ki67 was determined, or if there were differences between the two sites. The paper would be strengthened by some discussion regarding how their analysis could be adopted into routine clinical practice (if at all). Some additional validation of their findings using data from a completely different hospital, using different MRI machines, and different radiologists doing the analyses would be the true test of whether their findings have true clinical benefit.
Reviewer 3 Report
The purpose of this work is to establish a non-invasive measure to predict prognosis of PCNSL patients by ADC histogram. Authors measured apparent diffusion coefficient (ADC) histogram values and attempted to establish a univariate and multivariate Cox proportional hazard models to explore the possibility of whether ADC values could affect Ki 67 expression, PFS and OS. They found ADC values were significantly associated with OS and Ki 67 expression but not PFS. The overall experiments are well designed, and the data are basically supportive to the conclusion. However, as authors pointed out, ADC values have been studied to correspond to tumor markers such as 59 Ki-67, predict survival, and serve as an effective means of monitoring treatment response in PCNSL. Thus, the novelty of current study is unclear. Additional concerns are raised on the data presentation which demonstrate hardly a clear endpoint for clinical application.
Major concerns:
All of data presentation are not well prepared. For instance, the data shown in Table are mainly the raw data and confusing which should be organized and graphed before submission. There is no label on the graph of Fig. 1. It is not clear how this corresponding ADC histogram could be applied for clinic data. The advantage of whole tumor segmentation needed to be described. The novelty of this work, based on the cited publications, needs to be clarified. More data between PLWH and immunocompetent patients should be given as the PLWH is more susceptibility than normal person. In addition to Ki 67, more markers should be included such as the proliferation markers or markers associated with apoptosis or anti-apoptosis. In addition to hemorrhage, are there any factors affecting ADC values?Other informative tumor biomarker such as serum may be included.
Reviewer 4 Report
The authors elucidate possible benefit of ADC histogram analysis in cerebral lymphomas based upon a large patient sample. A histopathology correlation was performed with proliferation index Ki 67 and a clinical correlation with survival.
The histogram analysis is a contemporary tool.
The authors conclude that ADC histogram analysis can predict proliferation index and patient survival.
However, there are some limitations of the present study to address.
In the present study a 3 T and 1.5 T scanner was used. There is no statement, which system was used more and no sub group analysis was performed according to tesla strength. It is a well known fact that ADC values differ between tesla strenghts, manufacturer.
Of note, the same b-values 0 and 1000 s/mm² were used to harmonize the patient sample.
Please specify the VOI measurement. Did you draw the ROI within the whole contrast enhancement tumor area on t1-weighted images? Did you include the hemorrhage parts in your VOI? It is well known that hemorrhage can have a large influence on ADC values due to susceptibility.
Your explanatory figure lacks different sequences. In figure 1 you display the ROI on a presumably b0 image. Please add b1000 and ADC map, as well as the corresponding t1-weighted image for demonstration purposes.
Moreover, please add into the figure legend the ADC values of the patient.
You did not perform a interreader agreement for your ADC values, although 2 neuroradiologists performed the ROI measurements. Please add it into the study, at least for a smaller subset of the patients. It was shown that different ADC histogram parameters have different interreader agreements
For your histogram parameters you used 1th and 99th percentile, which might be almost equal to the ADCmin and ADCmax. You did not calculate entropy, which might be a promising ADC histogram parameter.
Please provide a table, which summarizes mean values of your ADC histogram parameters.
One big concern is the histopathology correlation.
There is no explanation of the Ki 67 calculation. Presumably, it is performed by bioptic specimen before any form of treatment. Please add it into the materials and methods part.
Moreover, your bioptic specimen might not be fully representative of the whole tumor. Please add this into your limitation part, as you compare different tumor areas. There is no possibility to retrospectively measure only the tumor part, which was acquired for histopathology?
You state that your results with a positive correlation of ADC values with Ki 67 index agree with the literature. However, this is a wrong statement. A higher Ki 67 index indicates a more malignant tumor, which might be reflected by MRI with a low ADC value. As was stated by Surov et al. in a meta analysis, a moderate inverse correlation exists between Ki 67 and ADC values (Oncotarget. 2017 Aug 24;8(43):75434-75444. doi: 10.18632/oncotarget.20406). This was also reported by Schob et al. 2016 in a small case series for CNS lymphomas.
In a recent meta analysis the pooled correlation coefficient between ADC values and KI 67 was r = -0.25 (95% CI, -0.53 to 0.04) (meyer et al., Clin Lymphoma Myeloma Leuk. 2019 Jun;19(6):e266-e272) using all lymphoma imaging studies.
Regarding your survival analysis only weak associations were identified. In table 3 the CI for kurtosis ranged from 0.58-4.9, which indicates no statistical significance. How was the p-vale of 0.03 obtained? Similar for ADCmin in HIV+ group.
In the discussion part, add a part of relationships between ADC values and celullarity, which might cause your findings.
Round 2
Reviewer 1 Report
Thank you for answering questions and improving the manuscript.
However, the question of using using ADC values from different scanners with different magnetic field strength is a questionable practice. The provided reference "J Comput Assist Tomogr. 2015 Sep-Oct;39(5):760-5." is a good argument, but ref. "Quant Imaging Med Surg. 2016;6(4):374–380. doi." states that ADC values are depend on b-value thresholding. By reviewr's opininon that is still an open question and may or may not affect authors findings. Reviewer is left the judgment to the scientific community.
Reviewer 2 Report
There has been effort made to improve the readability of this manuscript.
My opinion is that it would read much better if much of the methods section (eg. MRI analysis, figure 1, patient descriptions) was written into the results prior to the presentation of all the data tables as this provides the context for the analysis.
Reviewer 3 Report
The authors have addressed some concerns raised which improves the clearance for the data and conclusion.
Reviewer 4 Report
Thank you for the opportunity to review the revised form the paper.
The authors improved the quality of the paper substantially.
The conclusions drawn by the results are still overambitious.
